# Empowering the Potential of CAR-T Cell Immunotherapies by Epigenetic Reprogramming

**DOI:** 10.3390/cancers15071935

**Published:** 2023-03-23

**Authors:** Maria Alvanou, Memnon Lysandrou, Panayota Christophi, Nikoleta Psatha, Alexandros Spyridonidis, Anastasia Papadopoulou, Evangelia Yannaki

**Affiliations:** 1Hematology Department-Hematopoietic Cell Transplantation Unit, Gene and Cell Therapy Center, George Papanikolaou Hospital, 570 10 Thessaloniki, Greece; 2Bone Marrow Transplantation Unit, Institute of Cell Therapy, University of Patras, 265 04 Rio, Greece; 3Department of Genetics, Development and Molecular Biology, School of Biology, Aristotle University of Thessaloniki, 570 10 Thessaloniki, Greece; 4Department of Medicine, University of Washington, Seattle, WA 98195-2100, USA

**Keywords:** CAR-T cells, epigenetics, immunotherapy, T-cell persistence, exhaustion, memory T cells, tumor infiltration, epigenetic reprogramming

## Abstract

**Simple Summary:**

Epigenetic mechanisms are key players in many diseases, including cancer. Unlike genetic changes, epigenetic modifications are reversible without leaving a permanent mark on DNA. Epigenetic therapies, using epigenome-influencing techniques, aim to normalize DNA methylation patterns or post-translational modifications on histones, ultimately reversing a malignant phenotype. Chimeric antigen receptor T cells (CAR-Ts) have revolutionized our therapeutic approach to cancer; however, several challenges need to be overcome that are currently limiting a broader application of CAR-T cell therapy. Epigenetic remodeling of CAR-T cells may unleash their potential by diminishing exhaustion, improving trafficking and penetrating capacity, and promoting the memory phenotype, ultimately resulting in increased CAR-T cell persistence and improved outcomes.

**Abstract:**

T-cell-based, personalized immunotherapy can nowadays be considered the mainstream treatment for certain blood cancers, with a high potential for expanding indications. Chimeric antigen receptor T cells (CAR-Ts), an ex vivo genetically modified T-cell therapy product redirected to target an antigen of interest, have achieved unforeseen successes in patients with B-cell hematologic malignancies. Frequently, however, CAR-T cell therapies fail to provide durable responses while they have met with only limited success in treating solid cancers because unique, unaddressed challenges, including poor persistence, impaired trafficking to the tumor, and site penetration through a hostile microenvironment, impede their efficacy. Increasing evidence suggests that CAR-Ts’ in vivo performance is associated with T-cell intrinsic features that may be epigenetically altered or dysregulated. In this review, we focus on the impact of epigenetic regulation on T-cell differentiation, exhaustion, and tumor infiltration and discuss how epigenetic reprogramming may enhance CAR-Ts’ memory phenotype, trafficking, and fitness, contributing to the development of a new generation of potent CAR-T immunotherapies.

## 1. Introduction

Advances in cell and gene engineering in the last decades, along with a deeper understanding of the immune system’s role in controlling cancer, have revolutionized treatment strategies for previously incurable malignancies [1]. Among adoptive cell therapies, chimeric antigen receptor T cells (CAR-Ts) hold the lead, due to their remarkable clinical success against relapsed, refractory B-cell malignancies [2]. CAR-Ts are T cells genetically engineered to express a synthetic CAR, an anti-cancer monoclonal antibody anchored into the cell membrane and spliced to intracellular T-cell receptor (TCR) signaling domains. This antibody-based extracellular receptor structure alters T-cell capacity for immune surveillance by redirecting their specificity to surface antigens of cancer cells. CAR-Ts demonstrated unprecedented potency against difficult-to-cure hematological cancers, which resulted in their approval by the US Food and Drug Administration (FDA) and the European Medicines Agency (EMA) for B-cell hematological malignancies [3,4,5]. Nevertheless, relapse or resistance, as well as the CAR-Ts’ overall poor performance in the solid tumor setting, represent limitations associated with their use that need to be overcome [6,7,8,9,10]. It has become clear that robust expansion and long-term persistence of CAR-T cells are key elements in achieving complete and durable responses. The persistence, however, of functional CAR-Ts is hampered by several factors, including T-cell exhaustion, differentiation, prolonged antigen stimulation, rejection of immunogenic CARs by the host-immune system, contraction when antigen concentration is insufficient to drive T-cell proliferation, activation-induced cell death, or replicative senescence [11,12,13] and many efforts have been made towards increasing the persistence of T-cell therapy products in vivo [14,15,16,17]. In this review, we outline the main characteristics of epigenetic regulation and describe the epigenetic modifications that drive T-cell persistence, with emphasis on those underlying T-cell exhaustion, differentiation, and infiltration. We also touch upon the epigenetic reprogramming of CAR-Ts and the potential to counteract T-cell exhaustion, promote stemness, and augment intrinsic T-cell trafficking mechanisms towards unleashing the full potential of CAR-T cell immunotherapy.

## 2. Epigenetic Modifications

The term “epigenetics” (Greek: epi-above; -genetics) was introduced in the late 1950s by Conrad Hal Waddington to describe inheritable changes in phenotype and gene function during cell division, not due to alterations in the DNA sequence but modifications of the chemical state of DNA structure and function.

Epigenetics plays an important role in the tight control of chromatin function and gene expression, in the form of DNA modifications (cytosine methylation or hydroxy methylation), histone modifications (acetylation, methylation, ubiquitination), non-coding RNA (ncRNA)-associated modifications, and higher-order-associated modifications [18].

### 2.1. DNA Modifications

DNA methylation usually refers to a methyl group bound to a cytosine (C) nucleotide on the 5′ end of a guanine nucleotide (G) in the DNA, at a cytosine-guanine pair (CpG). High density CpG islands are hypermethylated regions, regulated by three types of enzymes called DNA methyltransferases (DNMTs); DNMT1 preserves the methylation pattern post replication by methylating the new unmethylated strand, while DNMT3A and DNMT3B establish de-novo methylation [19].

Methylation of CpG islands near transcription start sites (TSS) or proximal promoter loci results in methylation of the promoter itself, often leading to suppression of gene transcription and decreased gene expression by directly impairing the binding affinity of DNA to transcription factors or by recruiting repressive histone-modifying enzymes. DNA methylation is not restricted to promoter sequences, though; enhancers and cis-regulatory elements interacting with the promoter may also be methylated. Enhancer methylation usually inhibits the binding of transcription factors, thus repressing or reducing gene expression. Genes can be methylated/demethylated during life and then deactivated/activated, respectively, or they can be permanently methylated [18,20] (Figure 1).

Such DNA modifications can be exogenously induced by pharmaceutical and non-pharmaceutical means. Quite recently, the CRISPR/Cas9 system, a widely used and easily programmable platform for precise genome editing, has emerged as a tool for targeted modification of DNA methylation. A catalytically inactive or “dead” Cas9 (dCas9), usually bearing the catalytic domain of an epigenetic effector, can bind to a DNA target sequence of either promoter or enhancer regions or both [21] and alter epigenetic marks, thus leading to gene activation or repression [18,21,22].

### 2.2. Histone Modifications

Histone proteins, especially H2A, H2B, H3, and H4, which form the core nucleosome complex, undergo many post-translational epigenetic modifications. Methylation, acetylation, and ubiquitination are considered the major histone epigenetic modifications.

Histone methyltransferases (HMT) catalyze the transfer of methyl groups to lysine (K = Lysine) or arginine (R = Arginine) residues of histone (H3 and H4) proteins [23]. Trimethylation of lysine 9 or lysine 27 of histone H3 (H3K9me3/H3K27me3) or lysine 20 of histone 4 (H4K20me3) is prevalent in heterochromatin, priming transcriptional suppression [24]. In contrast, H3K4, H3K36, and H3K79 are present in euchromatin and thus correlate with active transcription [25]. Histone modification was thought to be permanent until the discovery of the lysine-specific histone demethylase 1A (LSD1 encoded by the KDM1A gene), which demethylates H3K4 and H3K9, leading to a dynamic remodeling of chromatin [24]. The LSD1 gene has been observed to be upregulated in T-cell acute lymphoblastic leukemia (T-ALL) and many other types of cancer [26]. It has also been shown that demethylases KDM6A (Utx) and KDM6B (Jmjd3) contribute to form chromatin structure and regulate gene expression by reversing repressive histone alterations [27] (Figure 1).

Histone acetylation (histone 3 of lysine 27 [H3K27ac]) is regulated by histone acetyltransferases (HATs) and histone deacetylases (HDACs), playing a crucial role in chromatin remodeling and regulation of gene transcription [28]. HATs are epigenetic enzymes that install acetyl groups into lysine residues associated with transcription activation, while HDACs remove acetyl functional groups from lysine residues of cellular proteins being associated with more condensed chromatin and transcriptional gene silencing [28,29] (Figure 1).

Ubiquitination is a reversible epigenetic histone modification, the status of which is determined by two enzymatic activities involving addition and removal of the ubiquitin moiety from histones, and it plays a fundamental role in shaping the chromatin epigenetic landscape and cellular identity [25]. The ubiquitin modification can be removed by a group of specialized deubiquitinating enzymes (DUBs) that hydrolyze the ubiquitin/histone substrate peptide bond. Ubiquitination is usually observed in H2A at lysine 119 (H2AK119ub1) and H2B at lysine 120 (H2B K120ub1). H2A ubiquitination is associated predominantly with transcriptional repression and considered a repressive mark, whereas H2B ubiquitination appears to be involved both in transcriptional activation and gene silencing [30,31]. (Figure 1).

### 2.3. Non-Coding RNA-Associated Modifications

As it is known, most transcripts in humans may not encode for proteins, but still play crucial roles in cell differentiation and function. There are two main groups of regulatory non-coding RNAs (ncRNAs) based on their size: short chain ncRNAs (small nuclear RNAs (snRNAs), micro-RNAs (miRNAs), silencing RNAs (siRNAs), small nucleolar RNAs (snoRNAs), ribosomal RNAs (rRNAs), transfer RNAs (tRNAs), circular RNAs (cRNAs), and piwi-interacting RNAs (piRNAs)) and long ncRNAs [32]. It has been shown that especially miRNAs and siRNAs are able to alter histone deacetylation and methylation, or DNA methylation, and thus effectively silence genes [33,34]. The miRNAs are naturally occurring double-stranded RNAs (dsRNAs), approximately 22 nucleotide bases long, that suppress specific genes by RNA-induced silencing at the post-transcriptional level, although they can also increase the expression of a target mRNA [35,36]. In addition, miRNAs contribute to several biological processes, including cell proliferation, apoptosis, cell death, and differentiation [37]. The miRNAs can act as epigenetic modulators per se by targeting epigenetics-associated enzymes such as DNA methyltransferases (DNMTs), deacetylases (HDACs), and histone methyltransferases (EZH), whereas their expression can also be epigenetically regulated by DNA methylation and RNA or histone modification [38,39]. The miR-148 targets the methyltransferases DNMT3B and DNMT1, leading to inhibition of cell proliferation and increased apoptosis [40]. The miR-449a promotes apoptosis and differentiation through targeting HDAC1 [41,42]. Other mi-RNAs, such as miR-152, miR-185, and miR-342, lead to DNA hypomethylation, promoting the expression of tumor-suppressor genes via DNMT1 [43].

The miRNAs are commonly found dysregulated in human cancer, where the aberrant patterns of miRNA expression are majorly associated with epigenetic alterations [44], suggesting that miRNAs could be important targets for epigenetic cancer therapy. For instance, in non-small-cell lung cancer (NSCLC), the (down)regulated expression of the miR-29 family (29a, 29b, and 29c) that normally exhibits a tumor suppressive potential and is associated with a poor prognosis is inversely correlated with the (over)expression of the DNA methyltransferases DNMT3A and DNMT3B, which can be directly targeted by miR-29s. By enforcing miR-29’s expression in lung-cancer cell lines, normal patterns of DNA methylation are restored, and methylation-silenced tumor suppressor genes are re-expressed, leading to inhibition of tumorigenesis. In contrast to their epigenetic regulation, miRNAs can also target epigenetic regulators. Cancer-specific dysregulation (reduced expression or genomic loss) of several miRNAs (26a, 98, 124, 214, 101, 137, and let-7) can lead to aberrant histone modifications and appears to be one of the major causes of EZH2 overexpression in several types of cancer (prostate, breast, bladder, gastric, lung, and renal) [43] (Figure 1).

### 2.4. Epigenome Editing

Epigenome editing represents epigenomic modifications induced by genome editing approaches in which genome alterations are not part of the DNA itself but are inheritable during cell division, reversible, and non-permanent.

CRISPR epigenome editors consist of a catalytically inactivated “dead” Cas9 (dCas9) tethered to the catalytic domain of epigenetic effectors such as DNMTs or ten-eleven translocation (TET) methyl-cytosine dioxygenases, HATs, or HDACs to potentiate or repress gene expression. A gRNA complementary to the target DNA sequence navigates the CRISPR-dCas9 epi-editor to the target site, whether that is a promoter or distal cis-regulatory sequence [45]. Epigenetic editing can be used to promote (CRISPR activation, CRISPRa) or attenuate (CRISPR interference, CRISPRi), the transcriptional activity based on the recruitment of transcriptional activators (VP64) or repressors (Krüppel associated box (KRAB) domain) to specific sites [46,47]. Epigenome editing for transient expression of transcriptional repressors, such as DNMT3A, or combinations of engineered transcriptional repressors (DNMT3a and KRAB domain), to the regulatory sequences of a gene of interest in primary T-lymphocytes induces repressive histone marks and de novo DNA methylation, establishing long-term memory of the repressive epigenetic state, which can be reverted only by targeted-DNA demethylation [48].

## 3. Epigenetic Reprogramming to Circumvent Challenges of CAR-T Cell In Vivo Performance

### 3.1. T-Cell Stemness and Memory

Immune memory is an essential feature of the immune system, which allows an organism previously exposed to foreign antigens to rapidly recognize a specific antigen and effectively eliminate it [49]. The predominant model is that during a natural T-cell response, naïve T cells encountering their cognate antigen, clonally expand and progressively differentiate towards both terminally differentiated effector T cells (T_EFF_) and memory T cells. T_EFF_ directly or indirectly destroy the target-antigen and then, in their vast majority undergo apoptosis while memory T cells persist and mediate effective surveillance [50,51], thus providing an immediate and enhanced response upon re-challenge [49]. Memory T cells are phenotypically and functionally heterogeneous, being subdivided into central memory (T_CM_, CD45RA^−^CCR7^+^CD62L^+^) cells and effector memory (T_EM_, CD45RA^−^CCR7^−^CD62L^+/−^) cells [52]. A special memory T-cell subgroup with superior proliferative and differentiation ability, the “stem-like memory T cells” (T_SCM_, CD45RA^+^CCR7^+^CD62L^+^CD27^+^CD28^+^IL-7R^+^CD95^+^) [53], in contrast to all other memory cell subsets requiring activation, originates from naive T cells (T_N_) in a resting state. Thus, the transition from naive T cells to effectors is represented by the following hierarchy: T_N_ → T_SCM_ → T_CM_ → T_EM_ → T_EFF_ [54,55]_._ Sustained immunity requires that CD8^+^ T cells have the capacity for long-lived persistence and effector function, long after their initial exposure to antigen (Figure 2).

It is now clear that CAR-Ts’ differentiation stage critically impacts their proliferation and survival, and certain immunophenotypes of the starting material or the final cell product have been associated with positive or adverse patient outcomes following adoptive CAR-T cell transfer [56,57]. In preclinical models, T_N_- and T_CM_-derived CAR-Ts outperformed their T_EM_-derived counterparts [56,58], while CD30-CAR-Ts generated from T_SCM_-like cells presented long-term persistence and fully eradicated the tumor, even after tumor rechallenge [59]. Recently, it was reported that an elevated frequency of T cells with memory-like characteristics, or CD8^+^CD45RO^−^CD27^+^cells (antigen-experienced lymphocytes that persist at a resting state, possessing properties of long-lived memory cells [60]), in the starting T-cell population of CLL or multiple myeloma patients, was associated with long-term remission and a better clinical outcome [61]. These findings highlight the importance of cell composition in the leukapheresis product, the selection of defined cell subsets before CAR-T generation [62] or manufacturing protocols leading to cell products with stem cell-like characteristics [63,64].

#### 3.1.1. Epigenetic Regulation of T Cell Differentiation

The elucidation of mechanisms by which immunological memory is formed, starting from a naïve T cell activated by its cognate antigen and giving rise to multiple distinct cellular states, is of critical importance. Several studies indicate that the transition from T_N_ cells into long-lived memory or short-lived terminal effector cell populations follows progressive epigenetic changes through DNA methylation or histone modifications under the orchestrated function of a variety of transcription factors (Figure 2).

At the DNA level, DNMT3A plays a critical role in T-cell effector fate decisions. Long-lived memory cells were shown to originate from a subset of T_EFF_ cells through de-differentiation and acquisition of repressive methylation marks at naïve cell-associated genes, such as *L-selectin (CD62L)*, *C-C chemokine receptor 7 (CCR7)*, *transcription factor 7 (TCF7)*, and demethylation marks at loci of defined effector molecules, such as the perforin gene (*Prf1)*, *granzyme genes,* and *Ifng* (Figure 2). The methylation status was erased by genetic disruption of DNMT3A, which promoted memory fate decisions by enhancing demethylation and faster re-expression of naïve cell-associated genes, thus accelerating memory cell generation [65]. In contrast, TET2, an epigenetic regulator of CD8^+^ T cell effector- *versus* memory-cell fate decisions, mediates DNA demethylation and subsequent upregulation of effector-associated genes, such as *Interferon-γ (IFN-γ)*, *C-C motif chemokine receptor 5 (CCR5)*, *granzyme B*, and *tumor necrosis factor (TNF)* [62]. TET2 knockout in mice with acute viral infection, resulted in increased DNA methylation at gene loci encoding transcription factors that favor effector T-cell differentiation, including *T-Box Transcription Factor 21 (TBX21*, *encoding T-bet)*, *PR domain zinc finger protein 1 (PRDM1)*, *B lymphocyte-induced maturation protein-1 (Blimp-1)*, *interferon regulatory factor 4 (IRF4),* and *RUNX family transcription factor 3 (Runx3),* while promoting the acquisition of a memory T-cell fate with a differentially expressed methylation pattern in TET2-deficient *versus* wildtype CD8^+^ T cells and provided superior control upon pathogen re-challenge [66] (Figure 2).

At the histone level, both repressive and active histone marks related to memory T-cell differentiation have been identified at certain loci in distinct subsets of antigen-experienced CD8^+^ T cells. In T_N_ and T_SCM_ cells, canonical loci and enhancer regions associated with immunological memory (*TCF7*, *lymphoid enhancer-binding factor 1 (LEF1)*, *forkhead box protein O1 (Foxo1)*, and its target *Kruppel-like factor 2 (Klf2)*) were shown to be enriched in active histone marks (H3K4me3) and depleted of inhibitory marks (H3K27me3), whereas genes and promoters of effector function-related molecules, such as *IFN-γ*, its regulator *TBX21*, *granzyme B,* and *perforin* were correlated with repressive histone marks [67,68]. In T_CM_ and T_EM_ subsets, there were progressively fewer activating H3K4me3 marks and higher repressive H3K27me3 marks at the same loci, implicating declining chromatin permissiveness during terminal-cell differentiation [68]. *Blimp-1* has been associated with effector T-cell functions by directly recruiting the repressive chromatin-modifying enzymes H3K9 methyltransferase and HDAC2 (histone deacetylase) to targeted promoters (*Il2ra* and *CD27* loci), thus promoting terminal T-cell differentiation through negative regulation of memory-associated T-cell genes [69,70]. The histone methyltransferase *SUV39H1* has also been associated with the differentiation of CD8^+^ T cells into effector cells and the silencing of the stem cell/memory gene expression program through H3K9me3 catalysis [71]. The above studies strongly suggest a dynamic chromatin remodeling after activation of T_N_ cells in promoter regions of memory- and effector-signature genes that coordinates multiple gene expression programs in various CD8^+^T-cell subsets (Figure 2).

A linear model of memory development in circulating T cells was also suggested by integrative analyses of genome-wide epigenomic profiles, and a list of candidate human functional epigenetic regulators of T-cell memory differentiation was developed. The list comprised transcription factors already known to control T-memory cells, such as *B-cell lymphoma 6 (BCL6)*, *E2F2*, and *RUNX3*, as well as new candidates, including the *aryl hydrocarbon receptor (AHR)*, *CAMP responsive element binding protein 1 (CREB1)*, *protein C-ets-1 (ETS1)*, *Fli-1 proto-oncogene ETS transcription factor (FLI1)*, *forkhead box P1 (FOXP1)*, *forkhead box J3 (FOXJ3)*, *nuclear factor erythroid 2-related factor 2 (NFEL2)*, *nuclear respiratory factor 1 (NRF1)*, *regulatory factor X3 (RFX3)*, and *zinc finger protein 161 (ZFP161)*. Some of these factors not only drive transcriptional profiles during memory transition but are also under epigenetic regulation themselves [72].

#### 3.1.2. Targeting Epigenetic Programs to Enhance CAR-Ts’ Long-Term Fate

The persistence of CAR-Ts in vivo, and by extension, their enhanced anti-tumor activity, has been associated with a less differentiated, central, and stem-like memory T-cell phenotype over the effector-cell phenotype. Therefore, preserving the memory potential of CAR-T products is a rational intervention to elicit durable clinical responses. Indeed, clinical trials have shown that the infusion of CAR-Ts expressing memory signatures resulted in favorable outcomes [56]. T cells from complete-responding patients with CLL after receiving CD19-directed CAR-Ts were enriched in memory-related genes, whereas T cells from non-responders demonstrated upregulation of effector differentiation programs. In addition, sustained remission was associated with an elevated frequency of CD45RO^–^CD27^+^CD8^+^ T cells in the leukapheresis product [73].

Likewise, epigenetic profiling, and in particular, assessment of the DNA methylation pattern in the cell product itself, has revealed specific epigenetic signatures associated with complete response and enhanced progression-free and overall survival [74]. Multiple studies have reported the global alteration of the epigenetic landscape upon memory differentiation and identified key epigenetic regulators of the progressive differentiation of memory T cells [72,75]. Youngblood’s group has recently shown that CAR-Ts infused in patients with acute lymphoblastic leukemia undergo genome-wide DNA methylation changes during antitumor response, including repression of stem-associated genes such as *TCF7* and *LEF1*, a transition towards effector function, reduced memory potential, and ultimately exhaustion [76]. In line with this study, in the largest clinically annotated molecular atlas of CAR-T therapy to date, *TCF7* and *LEF1* were nominated as key transcription factors driving naïve and memory T-cell states, whereas *PRDM1 (Blimp-1)*, *TBX21,* and *ZEB2* were nominated as driving T_EFF_ cell states [77]. Consequently, specific epigenetic profiles could serve as readouts for the prediction of clinical outcomes and the identification of patients who would benefit the most from CAR-T cell therapy, whereas the epigenetic reprogramming of CAR-Ts could be used as a tool to generate long-lived and more efficacious CAR-Ts.

Indeed, the genetic disruption of *DNMT3A* or *PRDM1* in CAR-T cells prevented methylation of several key genes that regulate human T-cell differentiation, including *TCF7*, *LEF1, and PRDM1,* and resulted in stem-like CAR-T cells that maintained their proliferative capacity and effector functions despite prolonged antigen stimulation, ultimately translating to enhanced T-cell persistence and improved superior tumor control [78,79]. The magnitude of the translational potential of epigenetic reprogramming in determining T-cell fate is highlighted in the case of a patient with chronic lymphocytic leukemia who achieved remission after CD19-CAR-T infusion, in whom inadvertent biallelic disruption in a single clone of the *TET2 gene* that regulates DNA demethylation occurred, leading to massive expansion of this clone, represented by 94% of circulating CAR-Ts [80].

Finally, there are many non-coding RNA-mediated mechanisms with an impact on T-cell memory. Chen et al. demonstrated that miR-150 negatively regulates CD8 T-cell memory in vivo by targeting the c-Myb-Bcl-2/Bcl-xl survival circuit [81]. Ban and colleagues reported that miR-150 deficiency skewed CD8^+^ T cells into the TCM and TEM phenotypes rather than the effector T-cell (TE) phenotype in an acute infection model and improved the production of effector cytokines. In addition, miR-150 deficient memory T cells proliferated more robustly than their wild-type counterparts and displayed an enhanced recall response and improved protection against infections [82].

Overall, targeting epigenetic programs, could globally alter the differentiation status of CAR-Ts, enhance their stemness, and ultimately improve their efficacy (Major epigenetic regulators associated with T-cell differentiation are shown in Table 1*).*

### 3.2. Exhaustion

T-cell exhaustion is induced by excessive antigen signaling and represents a biological self-defense mechanism that disrupts the excessive immune response during chronic infections or autoimmune diseases [123]. T-cell exhaustion is a state of dysfunctionality characterized by the gradual loss of effector functions and functional unresponsiveness due to persistent exposure to antigenic stimuli [124], resulting in impaired elimination of viral and tumor antigens. Initially described in the setting of chronic viral infections, T-cell exhaustion has also been recognized in cancer, where prolonged exposure to antigen and the immuno-suppressive tumor milieu can lead to loss of effector function and sustained inhibitory receptor expression, thus jeopardizing the efficacy of CAR-T cancer immunotherapy. The exhaustion features comprise a sustained expression of several inhibitory receptors, such as *programmed death-1 (PD-1)*, *cytotoxic T-lymphocyte antigen-4 (CTLA-4)*, *lymphocyte-activation gene 3 (LAG3)*, *T-cell immunoglobulin domain 3 (TIM3)* [125], and an impaired ability to produce cytokines such as *IFN-γ*, *TNF-α*, and *interleukin-2 (IL-2)* [126,127,128] (Figure 2).

Various studies suggest that T-cell exhaustion represents a pivotal hurdle for successful immunotherapy with CAR-Ts. At present, the mainly autologous source for CAR-T-cell immunotherapy is reasonably considered to be associated with shortcomings as the starting material for manufacturing is functionally impaired T cells derived from the tumor microenvironment, which may be exhausted already. Fraietta et al. observed that higher expression of exhaustion markers (PD-1, TIM3, and LAG3) in the CAR-T cell product correlated with poor responses in chronic lymphocytic leukemia patients treated with tisagenlecleucel (tisa-cel) [73]. In agreement with these findings, either a T-cell exhaustion signature or high proportions of circulating LAG3+ T cells were associated with poor molecular responses by cell-free DNA sequencing at day seven after infusion of autologous axicabtagene ciloleucel (axi-cel) or rapid disease progression after tisa-cel therapy in patients with large B-cell lymphoma, respectively [129]^.^

The sustained and high-level antigenic stimulation may be a key factor in the process of T-cell exhaustion in vivo [130]. Apart from antigen-dependent receptor signaling, antigen-independent receptor signaling induced by dimerization and (self-) aggregation of CARs that drives auto-activation, known as *tonic signaling*, has been implicated in driving T-cell exhaustion accompanied by a reduction in anti-tumor activity and poor CAR-T performance [131,132,133,134] (Figure 2). Recently, however, Singh et al. proposed that tonic signaling in certain contexts, in particular for CD-22-CAR constructs containing a 4-1BB costimulatory domain rather than CD28, can result in enhanced in vivo potency and CAR-T cell persistence depending on the single-chain variable fragment linker length (short vs. long) of the CAR [135], providing insights into the context-dependent functionality of CAR-Ts.

#### 3.2.1. Epigenetic Regulation of T-Cell Exhaustion

Epigenetic regulation is a critical mediator of T-cell exhaustion, which is associated with a heritable epigenetic imprint, distinct from effector and memory T cells. As it was demonstrated in chronic lymphocytic choriomeningitis (LCMV), T-exhausted (Tex) effector and memory virus-specific cells over their functional counterparts, which eventually lost the ability to elicit effector functions after sustained exposure to viral antigen, indicating that exhaustion is highly associated with extensive changes in chromatin accessibility [136]. Importantly, when Tex cells from chronically LCMV-infected mice were infused into infection-free mice and their functional, transcriptional, and epigenetic transition toward T-effector cells was analyzed, it was demonstrated that Tex cells, although they had acquired some features of memory T cells, were lacking the ability to proliferate and respond to a new infection, leading many to suggest that exhaustion is a cell fate decision [137,138].

The irreversible epigenetic imprint on Tex cells was further confirmed after the administration of immune checkpoint blockade therapy, such as anti-PD-1, its ligand (PD-L1), or anti-CTLA-4, as a means to rejuvenate T cells and increase their survival and proliferation [139]. Indeed, although PD-L1 blockade has been shown to reinvigorate Tex cells in a mouse model of chronic LCMV infection, it failed to remodel their epigenetic state, thus resulting in their subsequent re-exhaustion in the presence of high antigen concentration [139], suggesting that the exhaustion of Tex cells is fundamentally regulated by epigenetic mechanisms—rather than just by the overexpression of immuno-inhibitory receptors—and thus shaping an inflexible Tex fate.

The molecular signatures of the Tex state are still under investigation. By whole-genome bisulfite sequencing of antigen-specific murine CD8^+^ T cells at the effector and exhaustion stages of an immune response. Ghoneim et al. demonstrated that the effector-to-exhaustion transition of T cells is DNMT3A-mediated [140]. Once established, the de novo methylation programs restricted T cells’ rejuvenation and expansion potential during PD-1 blockade treatment; yet, blockade improved T-cell responses and tumor control during PD-1 immunotherapy [140]. Several transcription factors, such as *TCF7*, *TBX21, and Eomesodermin* (*Eomes*, a T-box transcription factor), known to regulate immune-related pathways, were identified as putative regulators of the DNMT3A-targeted loci [140]. Other studies identified the nuclear DNA-binding factor *TOX (thymocyte selection-associated HMG-box protein),* inducing Tex-specific epigenetic opening of an enhancer upstream of the *PDCD1* gene encoding PD-1, and the nuclear receptor transcription factors *NR4A* as central contributors in the epigenetic remodeling during T-cell exhaustion [100,141,142,143] (Figure 2). Belk et al. methodically performed genome-wide CRISPR screens in murine and human tumor models in order to develop an extended atlas of regulators of T-cell exhaustion, demonstrating an enrichment of epigenetic factors [144]. Among other already known regulators, e.g., Gata3, new epigenetic factors that contribute to chromatin and histone remodeling were also uncovered, including the canonical BRG1/BRM-associated factor (cBAF) family. In particular, *Arid1a* depletion was essential in exhaustion-associated chromatin remodeling that occurs during chronic antigen stimulation, finally leading to [144] improved T-cell persistence and tumor control.

Overall, the failure of core memory epigenetic circuits to recover, accompanied by the persistence of a largely open chromatin texture, which has been described as epigenetic “scarring” [145], highly supports the development of epigenetic remodeling interventions that increase the epigenetic plasticity and reverse exhaustion of Tex cells in T-cell immunotherapies.

#### 3.2.2. Targeting Epigenetic Programs to Overcome CAR-Ts’ Exhaustion

Exhaustion may lead to CAR-T languishing and inevitably to relapse after CAR-T therapy. Formulating strategies to slow down, prevent, or even reverse CAR-T exhaustion seems crucial for enhancing the treatment’s efficacy. Indeed, cancer immunotherapies aiming at Tex cell reinvigoration, such as monoclonal antibodies targeting PD-1, PD-L1, or CTLA-4, namely checkpoint inhibitors, alone or in combination with CAR-Ts, marked a breakthrough in cancer immunotherapy. Nonetheless, a plethora of patients are devoid of durable responses due to Tex cells’ epigenetic inflexibility.

To achieve an in-depth understanding of the mechanisms of exhaustion and the development of novel strategies to overcome exhaustion and improve CAR-Ts’ performance, epigenetics, and epigenome editing have greatly helped by offering a novel toolset for transcriptional down- or up-regulation.

PDCD1, the gene encoding PD-1, is transiently demethylated in activated T cells but remains transcriptionally demethylated in Tex cells, suggesting that the blockade of exhaustion-associated methylation programs may improve the clinical outcome after CAR-Ts (Figure 2). Indeed, knockout or small hairpin RNA (shRNA)-mediated knockdown of PDCD1 enhanced the anti-tumor efficacy of *C-type lectin-like molecule-1 (CLL-1)-*, *mesothelin-*, *epidermal growth factor receptor (EGFR)-*, *CD19-*, and *Glypican-3 (GPC3)-* CAR-Ts [95,96,97,98]. Genetic depletion or pharmacological inhibition of other exhaustion-mediating transcription factors or immuno-suppressive regulators such as the *hematopoietic progenitor kinase 1 (HPK1)*, driving T-cell exhaustion through the *HPK1-Blimp1* axis, has been shown to improve CAR-Ts’ function and consequently the immune responses in diverse mouse models of hematological and solid tumors [99]. Moreover, CRISPR-HPK1-edited CD-19 CAR-Ts are currently being tested in patients with relapsed or refractory CD19+ leukemia/lymphoma (NCT04037566). It has been recently suggested that transient cessation of CAR signaling using designs that incorporate periods of rest or inhibit proximal TCR or CAR kinases can restore functionality and induce epigenetic reprogramming in exhausted CAR-T cells. CAR-Ts treated with dasatinib, an Src kinase inhibitor, were rejuvenated to exhibit improved expansion, decreased inhibitory receptor expression, and functional reinvigoration due to the dasatinib-mediated inhibition of tonic CAR signaling that could induce rest and ultimately reverse exhaustion [146]. Alternatively, in the same context, CAR-T cell rest was induced by a drug-regulatable system enabling controlled CAR expression and tonic CAR signaling by the presence (ON) or absence (OFF) of a small molecule [146]. CAR-Ts in the OFF state exhibited diminished tonic signaling, a memory-like phenotype, and superior anti-tumor activity in vitro and following adoptive transfer into xenografts.

Given that DNA methylation, mainly by DNMT3A, can lead to exhaustion of CAR-Ts, targeting DNMT3A using approved DNA demethylating agents, such as decitabine and azacytidine, may improve their function by reversing exhaustion-associated DNA methylation programs. Indeed, it has been recently reported that CAR-Ts treated with the DNA methyltransferase inhibitor decitabine underwent DNA reprogramming, which remarkably enhanced their anti-tumor activity both in vitro and in vivo. Transcriptomic profiling revealed enrichment in genes associated with naive, early memory T cells, and non-exhausted T cells, as well as upregulation of immune synapse-related genes [147,148].

In addition, at the histone and chromatin level, Zhang et al. found that progressive loss of T-cell function during chronic viral infection was associated with decreased diacetylated histone H3 levels in both the virus-specific and total CD8^+^ cells and that treatment of Tex cells with histone deacetylase inhibitors restored diacetylated histone H3 and their functionality [149]. Another study identified *TOX* as a central regulator of transcriptional and epigenetic Tex cell programming and a major factor for the developmental inflexibility of Tex cells even after PD-1 blockade, thus suggesting that manipulation of TOX or TOX-dependent epigenetic paths by specific miRNAs or CRISPR/Cas9 editing may reverse CAR-Ts exhaustion and improve the clinical outcomes [100]. (*Major epigenetic regulators associated with T-cell exhaustion are shown in*
Table 1*).*

### 3.3. The Tumor Microenvironment

The tumor microenvironment (TME) consists, apart from tumor cells, of a broad variety of cell types, including endothelial cells, immune cells (lymphocytes and macrophages), stromal cells (fibroblasts), as well as non-cellular components of the extracellular matrix (ECM), including several growth factors and cytokines [150,151]. This complex and hostile TME supports the survival of cancer cells, their uncontrolled proliferation, and immune escape [152], while preventing T-cell infiltration and driving T-cell dysfunction, ultimately inhibiting the efficacy of T-cell immunotherapies (Figure 2).

The TME poses a number of physical barriers to T-lymphocyte trafficking towards the tumor, including cancer-associated fibroblasts (CAFs) and abnormal vasculature at the tumor site that block T-cell entry [152]. T cells also need to bypass or overcome various ongoing immuno-suppressive processes in the TME, such as the secretion of immuno-suppressive cytokines such as TGF-β, ligand signaling via inhibitory receptors such as PD-L1, or competition for nutrients within the TME [7]. Several reports have so far highlighted the negative impact of TME on T-cell function. For instance, myeloid-derived suppressor cells (MDSCs), a heterogeneous cell population present at very low frequencies in healthy subjects, accumulate during inflammation and cancer. MDSCs with increased *indoleamine 2,3-dioxygenase (IDO)* activity are more common in chronic lymphocytic leukemia (CLL) patients, suppressing the activation of T cells and inducing suppressive Treg cells. Interestingly, CLL cells can induce the conversion of monocytes from healthy subjects into MDSCs, suggesting the existence of a regulatory interconnection between CLL cells, MDSCs, and Tregs [153]. Likewise, the expression of the inducible *T-cell costimulator ligand (ICOSL)* on acute myeloid leukemia (AML) cells, provided co-stimulatory signals for the expansion of ICOS + Tregs, which, through the secretion of IL-10 and TGF-β, led to immune evasion, further promoting the proliferation of AML cells [154]. In patients with refractory B-cell lymphoma receiving CD19-CAR-Ts, tumor-associated macrophage infiltration was inversely correlated with remission, implicating that the presence of macrophages in the TME impairs CAR-Ts’ function [155] (Figure 2).

In the context of solid tumors, besides its immuno-suppressive features, TME further restrains CAR-Ts ability to efficiently migrate to and penetrate the tumor parenchyma [156], posing a far more complicated challenge to tackle. Immune cell trafficking to tumors is spatiotemporally supervised by a coordinated rolling and adhesion of circulating T cells on the endothelium, leading to extravasation and subsequent infiltration of the malignant tissue. The interaction of T cells with the endothelium is based on chemokines and chemokine receptors (CCR); successful extravasation requires the acquisition of highly specialized T-cell homing receptors to determine the type of immune cells to be recruited to the TME and facilitate T-cell migration through sensing chemoattractant gradients [157,158]. Mismatches between the tumor chemokines and cognate receptor expression, tumor-induced aberrations of endothelial vessels and adhesive molecules, immunoediting of tumor antigen expression, immunosuppression, and recruitment of cancer-associated fibroblasts [158,159] represent major obstacles preventing T-cell homing and penetrance to the tumor (Figure 2).

#### 3.3.1. Epigenetic Regulation of T-Cell Infiltration

Epigenetic changes in the TME play a central role in tumor initiation, progression, and spreading and can occur in response to both intrinsic mechanisms and TME—T-cell interactions. Indeed, genome-wide DNA methylation analysis of tumor-infiltrating and circulating CD4^+^ T cells from glioblastoma (GBM) patients revealed unique DNA methylation and gene expression patterns between T cells derived from the different sources, as regards genes implicated in T-cell activation, aggregation, and chemotaxis, suggesting that the glioblastoma microenvironment may hamper the anti-tumor response by inducing significant epigenetic alterations in tumor-infiltrating T cells [160].

Moreover, complex interactions between T cells and other cellular components of the TME can lead to suboptimal tumor infiltration. Remarkably, in murine pancreatic cancer models, Borgoni et al. [161] demonstrated that tumor-infiltrating T cells are epigenetically shaped by the microenvironment and in particular by tumor-associated macrophages (TAMs) towards a pro-tumoral phenotype. By assessing the epigenetic profile of T cells obtained from mice implanted with pancreatic ductal adenocarcinoma (PDA) cells and either treated with a cytotoxic agent that selectively ablated TAMs or left untreated, it was found that in the absence of TAMs, the TME was highly populated with epigenetically remodeled *IL-10*, *T-bet,* and *PDCD1* promoters, CD4^+^ and CD8^+^ T-cell fractions, forming an anti-tumor phenotype [161]. The latter implicates that strategies deploying epigenetic modulators may also be considered therapeutic for pancreatic adenocarcinoma.

In another prime example of how TME interactions can coordinate complex molecular pathways to protect tumor elimination by immune cells, the cell-cell contact-dependent interaction of bone marrow mesenchymal stromal cells (BMSCs) with multiple myeloma (MM) cells led to the upregulation of *survivin (BIRC5)*, the anti-apoptotic capsase-3 inhibitor, rendering myeloma cells resistant to the cytotoxic machinery of T- and NK-cells [162]. Interestingly, epigenetic mechanisms have been recently shown to contribute to *survivin* dysregulation in human cancers, including either aberrant hypo- or hyper-methylation of the *survivin* promoter or altered *survivin* protein translation or mRNA degradation by binding of miRNAs to the 3′-untranslated region (UTR) of *survivin* mRNA, thus justifying the investigation of survivin-targeted therapy for cancer treatment [163,164,165]. Specifically, the BMSC-induced drug resistance of MM cells has been associated with *miR-101-3p* downregulation, thus the *miR-101-3p-survivin* interaction can serve as a druggable target to potentially sensitize MM cells to anti-myeloma drugs [166].

Likewise, chemokine expression is regulated not only by cancer’s intrinsic genetic mechanisms and environmental cues but also by epigenetic effects in the tumor microenvironment, including histone modifications and DNA methylation. Indeed, in a primary ovarian cancer model, enhancer of *zeste homologue 2 (EZH2)*-mediated histone H3 lysine 27 trimethylation (H3K27me3), as well as DNMT1-DNA methylation at the promoter of *CXCL9* and *CXCL10*, suppressed the secretion of central T helper 1 (Th1) chemokines CXCL9 and CXCL10 by malignant cells, leading to inhibition of the trafficking of T_eff_ cells into TME [103]. In addition, the forced expression of *CXCL14*, another chemokine recognized as an important tumor suppressor gene that is epigenetically silenced during lung carcinogenesis, led to dramatic tumor growth reduction [167]. In response to an inflammatory challenge, IL-15 signaling drives *Gcnt1* expression, which is critical for the synthesis of 2 O-glycans modulating the rapid, memory (but not naïve) CD8^+^ T-cell trafficking to inflamed tissues or tumors in an antigen-independent manner. Memory and naïve CD8^+^ T cells exhibit an opposite pattern of epigenetic modifications at the *Gcnt1* locus and distinct trafficking patterns. In particular, the *Gcnt1* locus undergoes chromatin remodeling to an open configuration only in memory CD8^+^ T cells, which could be potentially manipulated in order to achieve efficient trafficking to inflamed tissues [168].

Collectively, it can be envisioned that these tumor intrinsic epigenetic networks can be genetically or pharmacologically targeted to augment T-cell trafficking to tumors, alter the immune phenotype of the tumor, or even reverse the anti-inflammatory properties of the TME, thus leading to tumor regression by potentiating the effect of adoptive cell therapies.

#### 3.3.2. Cell Metabolism and Epigenetics

Cellular metabolism plays a central role both in cancer cell growth and proliferation as well as in T-cell fitness and potency, and anti-tumor responses are potentially regulated by metabolic reprogramming. An interplay of metabolic events between T cells and the TME is taking place in the context of cancer, where metabolic and nutrient changes in the TME reshape the metabolic status of T cells, impacting their activation, differentiation, and exhaustion while generating an immuno-suppressive milieu. On the other hand, the cell metabolome is interrelated to the epigenome, and epigenetic modifications in cancer are strongly linked to cellular metabolism by controlling expression of enzymes involved in metabolic pathways, while metabolism affects epigenetic regulation through the biosynthesis of macromolecules and energy production [169,170,171,172]. These interactions are bidirectional and synergistic in cancer. For instance, histone lysine methyltransferase SETD2, involved in complex different histone modifications, links epigenetic reprogramming with “metabolic memory” in prostate cancer by contributing to the EZH2 and AMPK signaling pathways [169]. The c-Myc oncogene-mediated inhibition of succinate dehydrogenase complex subunit A (SDHA) via acetylation and activation of deacetylase degradation pathways leads to cellular succinate accumulation, further triggering H3K4me3 activation, tumor-specific gene expression, and, thus, tumor progression [170]. In addition, tumors with the H3.3K27M mutation could promote glutamine and glucose metabolism, leading to an epigenetic status marked by H3K27me3 deficiency, while the genetic (e.g., shRNAs) or pharmacological interruption of these metabolic/epigenetic pathways results in suppressed H3.3K27M cell growth in vitro and tumor progression in vivo [171,172].

Metabolic events involve a variety of metabolites, such as acetyl-CoA, nicotinamide adenine dinucleotide (NAD+), S-adenosyl methionine (SAM), ATP, flavin adenine dinucleotide (FAD), succinate, and α-ketoglutarate (αKG). These molecules, by dynamically regulating the metabolic status of DNA and histones, play essential roles (as substrates or cofactors) in epigenome control during cancer development ([169] *and references within*).

Apart from the effect of the metabolome/epigenome on tumor growth, their interplay has important consequences for T-cell differentiation, function, and fate [170,171]. Numerous, external (i.e., oncometabolites, cytokines) or internal metabolic factors (i.e., acetyl-CoA, SAM) can influence the epigenetic landscape of T cells, whereas epigenetic reprogramming directly influences T-cell metabolism by affecting regulatory enzymes involved in glycolysis, OxPhos, or mitochondrial biogenesis ([172] and references within). The broad use of multi-omics shed light on the interrelationship between epigenetics and metabolism and cultivated the idea of manipulating T-cell metabolism via epigenetic modifications towards improving cancer immunotherapy. The enhanced capacity for glycolysis of terminally differentiated CD8^+^CD28^−^ memory T cells, an accumulating population during human aging, has been linked to downregulation of the Sirtuin1‘ (SIRT1)/FoxO1 axis; empowering the SIRT1–FoxO1 axis has been proposed as a targeted intervention for reprogramming terminally differentiated memory T cells and delaying the immune aging process [169,173]. Pharmacologic inhibition of Sirtuin-2 (Sirt2), a NAD^+^-dependent deacetylase inhibiting T-cell metabolism and impairing T-cell effector functions, endows human tumor-infiltrating lymphocytes (TILs) with superior metabolic fitness and effector functions [174]. Protein arginine methyltransferase 5 (PRMT5) has a great impact on T-cell differentiation and function [175] through the induction of cholesterol biosynthesis, while specific Prmt5 deletion in CD4^+^ T cells suppressed Th17 cell differentiation and protected mice from developing experimental autoimmune encephalomyelitis [176]. The reported metabolic exhaustion of TILs has been attributed to direct competition between TILs and cancer cells for metabolic resources and subsequent environmental stress-induced epigenome remodeling within TILs, resulting from the loss of histone methyltransferase EZH2. In contrast, reprogrammed T cells expressing a gain-of-function EZH2 mutant displayed an enhanced ability to inhibit tumor growth in vitro and in vivo, thus suggesting that manipulation of T-cell EZH2 in cellular therapies may provide cellular products able to withstand solid tumor metabolically deficient environments [177,178]. The metabolic reprogramming of effector CD8^+^ T cells through protein kinase MEK1/2 inhibition enhanced mitochondrial biogenesis and fatty acid oxidation and induced T_SCM_ with increased self-renewability, multipotency, and proliferative capacity, less exhaustion, and strong antitumor effects in vivo [179,180].

MiRNAs also affect T-cell metabolism. Forced expression of miR-155 in tumor antigen-specific T cells improved tumor control, and the miR-155-transduced T cells exhibited increased proliferation and effector functions associated with higher glycolytic activity independent of exogenous glucose, thus implicating that miR-155 may optimize the antitumor activity of adoptively transferred TILs by rendering them more resistant to the glucose-deprived environment of solid tumors [112,181,182]. The miR-143 overexpression enhanced the specific killing of HER2-CAR-T cells against esophageal cancer by promoting memory T-cell differentiation and metabolism reprogramming (T-cell glucose uptake and glycolysis inhibition) through glucose transporter 1 (Glut-1). The miR-143 expression and the regulation of T-cell differentiation are suppressed by IDO and its metabolite, kynurenine, in the tumor microenvironment, so IDO inhibition in the TME might increase the expression of miR-143 and enhance the antitumor effects of T cells by promoting T-cell differentiation [88]. Decreased miR-34a expression in hypoxic tumor environments has been correlated with increased lactate concentration and increased LDHA in lactate-abundant tumors, resulting in impaired T-cell immune surveillance, pointing out T-cell LDHA and miR34a as potential therapeutic targets for improved adoptive immunotherapy [183].

#### 3.3.3. Targeting Epigenetic Programs to Alter the TME and Enhance CAR-Ts’ Infiltration

Epigenetic modulation of TME-resistant CAR-Ts may hold the key to unleashing their true potential, even within hostile TMEs. To overcome the hurdle of localization to the tumor and facilitate T-cell migration towards the malignant milieu, the so-called armored CAR-Ts, which, in addition to targeting tumor antigens, constitutively secrete T-cell stimulating cytokines (such as IL-12,-15,-18, -21), have been generated to enhance the CAR-T cells’ homing to and activity against solid tumors (as reviewed in [184,185,186]).

In a recent work by Zou F. et al. triple ablation of inhibitory receptors *PD-1*, *TIM-3,* and *LAG-3* by short hairpin RNA (shRNA) in anti-Her2 CAR-Ts resulted in their epigenetic reprogramming and increased chromatin accessibility of the CD56 gene as well as key genes, including *IFN-γ, TNF-α,* and *Bcl-2;* their enhanced transcriptional expression in combination with transcriptional upregulation of chemokines including CXCL9, CXCL10, CXCL12 led to enhanced CAR-T infiltration into the tumor and thus superior disease control in murine models [101].

Moreover, Ding ZC et al. demonstrated that co-expression of a *constitutively active form of signal transducer and activator of transcription 5* (CA-STAT5) and a CD19 CAR enabled extensive epigenetic and chromatin remodeling in tumor-specific CD4^+^ cells, which diverted the fate of CD4^+^ cells from exhaustion to polyfunctionality and gave rise to tumor-tropic, anti-tumor T cells capable of vigorously accumulating within sites of lymphoma and eliciting anti-tumor CD8^+^ T-cell responses with a high cure rate in mice with advanced lymphoma

Furthermore, approaches utilizing pharmacological hypomethylation in the form of DNMT inhibitors (DNMTi) have shown that decitabine-treated CAR-Ts (dCAR-Ts) possess superior effector function and are able to secrete high levels of chemokines such as CXCL1, CXCL8, CXCL9, CCL1, and CCL3 by modifying DNA methylation programs. RNAseq analysis of dCAR-Ts revealed not only higher memory-associated and relatively lower exhaustion-associated gene expression but also increased leukocyte chemotaxis and lymphocyte migration, thus improving the homing ability to the tumor and facilitating the eradication of bulky tumors in in vivo models [148]. In cancer cells treated with decitabine or other epigenetic drugs inducing DNA demethylation and/or histone acetylation, the repressed production of T helper 1 (TH1)-type chemokines *CXCL9* and *CXCL10* by the tumor can be reversed, leading to increased infiltration of CD8^+^ T cells to tumor sites, arresting the tumor progression, and improving the outcome of adoptive T-cell immunotherapy in tumor-bearing mice [103]. Epidrugs can also upregulate the Ag-processing machinery and increase neoantigen presentation by MHC class I in tumor cells. Indeed, in immunologically “cold” tumors, such as glioblastoma (GBM), decitabine was shown to increase *neoantigen-* and *cancer testis antigen (CTA)*-specific T-cell activation through DNA hypomethylation, leading to enhanced antigen-specific T-cell-mediated toxicity to decitabine-treated cancer cells and thus being implicated as a sensitizing agent for immunotherapy [187]. Likewise, *lysine-specific histone demethylase 1A (LSD1)* ablation by CRISPR/Cas9-mediated disruption led to double-stranded RNA (dsRNA) stress and activation of type-1 interferon, thus stimulating tumor immunogenicity, anti-tumor T-cell immunity, and CD8^+^ T-cell infiltration. Moreover, *LSD1* ablation overcame the resistance to anti-PD-1 therapy in a model of checkpoint blockade-refractory mouse melanoma, suggesting LSD1 inhibition in combination with PD-(L)1 blockade as a novel cancer treatment [188].

The *CXCR3* axis is a critical pathway for immune cell recruitment to solid tumors that can be harnessed to increase anti-tumor responses. The increased expression of *CXCR3* ligands by the tumor cells, or of *CXCR3* on T cells, enhances T-cell trafficking towards the tumor. The expression of *CXCR3 ligands*, *CXCL9, and CXCL10*, is known to be epigenetically repressed in colon and ovarian tumor cells through the H3K27me3 activity of the *polycomb repressive complex 2 (PRC2)* and DNA methylation. By pharmacological inhibition of either the *PRC2 component EZH2* or *DNMT1,* it was possible to restore the tumoral *CXCR3* ligand epigenetic silencing and therefore enhance CD8^+^ T-cell trafficking and tumor growth control [103].

As stated above, cell–cell interactions between BMMSCs and myeloma cells protect the latter against the cytotoxic machinery of T- and NK-cells [189] by upregulating the anti-apoptotic *survivin*. This effect of immune escape is of relevance also to CAR-T immunotherapy, where it could be circumvented by designing CAR-T cells with high cytolytic capacities or by combining CAR-T cells with epigenetic inhibition of antiapoptotic proteins in MM cells [190].

In addition, the recruitment of various immuno-suppressive cells expressing T-cell inhibitory receptor ligands in the TME impairs tumor invasion by CAR-T cells and facilitates immune resistance and relapse. Reinvigoration of immune responses can be addressed by targeting key immune checkpoint regulatory networks. Indeed, GBM tumor cells treated with *EGFR CAR-T* cells soon acquired resistance to treatment and relapsed due to the upregulation of immuno-suppressive genes, including inhibitory immune checkpoints [191]. *BRD4*, an epigenetic modulator and member of the *bromodomain and extra terminal (BET)* subfamily of human bromodomain proteins, is required for the activation of these immuno-suppressive genes and, among other proteins, regulates PD-L1 [192]. Combination therapy with CAR-T cells and *BET inhibitors (*i.e.,* JQ-1)* suppressed PD-L1 and TIM-3 expression and tumor cell growth in GBM, AML, and ovarian cancer models, thus improving the efficacy of immunotherapy by both rescuing CAR-Ts from exhaustion and making the tumor mileu more hospitable to T cells [191,193,194]. (*Major epigenetic regulators associated with T-cell infiltration are shown in*
Table 1).

## 4. Concluding Remarks

CAR-T cell immunotherapy has led to impressive remission rates in patients with previously incurable B-cell hematological malignancies. Nevertheless, remissions can be brief in a substantial number of patients, while only rarely occurring in patients with solid tumors who remain largely refractory to the CAR-T approach. Therefore, challenges arise as regards the optimization of CAR design and manufacturing, the improvement of response rates, the durability of remissions, the reduction in toxicity, and the broader applicability towards including the difficult to treat with CAR-T cells, solid tumors.

Multi-omics analyses, including genomics, epigenomics, immunogenomics, transcriptomics, proteomics, and metabolomics from both tumor and non-tumor tissues at the bulk and/or single-cell levels, have broadened our understanding of the mechanisms that control T-cell fate determination, polyfunctionality, and resistance development, identified novel tumor targets, biomarkers with core epigenetic signatures, and pathways of resistance [195], as well as allowed patient stratification on the basis of the expected benefit from CAR-T cell therapy [145,195]. Such deep mechanistic insights revealed the role of epigenetic mechanisms in the functional properties and overall fitness of CAR-T cells. Specific epigenetic loci have shaped an epigenetic signature (EPICART), correlating with complete responses and increased event-free and overall survival in CAR-T cell recipients [74].

Several studies have been developed to enhance adoptive immunotherapy outcomes through epigenetic modifications. Epigenetic interventions during in vitro manufacturing of CAR-T cells, either by genetic modification or transient pharmacological inhibition of specific epigenetic genes or enzymes, have offered the opportunity to fine-tune the CAR-T functional state towards long-lived memory CAR-T cells without compromising their effector function. Following infusion, the epigenetic interventions aim to circumvent epigenetic and transcriptional changes, which considerably affect effector functions, exhaustion, and tumor infiltration. Epigenetic strategies for reprograming CAR-Ts, including DNMTs inhibitors, HDAC inhibitors, ncRNAs, or multiplex modifications, have allowed CAR-Ts to persist long-term following adoptive transfer, mitigate exhaustion, enhance trafficking to the tumor, and boost their therapeutic effectiveness. Challenging the notion that exhaustion is an epigenetically fixed state, CAR molecules that have been engineered to enable conditional transient inhibition of CAR expression and tonic signaling or have been subjected to transient exposure to a TKI that reversibly inhibits proximal TCR or CAR signaling, exhibited a global epigenetic remodeling leading to prevention or reversal of T-cell exhaustion in vivo, even in cell populations that had already acquired epigenetic features of exhaustion [139,146,196].

Recently, the role of T cells as accelerators of inflammation, a state of chronic, low-level inflammation in the elderly characterized by an increase in the levels of pro-inflammatory molecules in blood and tissues and a loss of protective immunity as a result of genetic, environmental, or stochastic factors, has been recognized [197,198]. Changes introduced by such factors and possibly contributing to inflammaging are molecularly recorded by epigenetic modifications, being heritable to subsequent cell generations and leading to the accumulation of cell- and immune-related senescence [199]. Given the high numbers of late middle-aged and older patients who are candidates for CAR-T cell therapy but experience multiple age-related conditions, a number of epigenetic modulation strategies and senolytic CAR-T cells (targeting senescence-specific antigens) have recently been developed to optimize CAR-T cell fitness for older patients and broaden CAR-T cell therapy to aging-related diseases, *as reviewed in* [200].

Nevertheless, the field is new, the epigenetic regulation is pleiotropic, and consequently, the intended CAR-T cell epigenetic remodeling is highly complex. Undoubtedly, epigenome editing integrated into the CAR-T manufacturing process will add to the production complexity and thereby increase the already high cost. In addition, epigenome editing technologies, although not relying on the alteration of the underlying DNA and thus considered presumably safer, should be highly faithful and extensively tested for as many currently underappreciated *on-* and *off-* target consequences as possible, given that the epigenetic regulators can affect multiple pathways within the cells and some of the candidate target genes (*DNMT3A*, *PRDM1)* are associated with tumorigenesis.

Overall, epigenetics has provided new opportunities for optimizing CAR-Ts and maximizing their clinical efficacy, as well as for broadening their therapeutic use to include patients with solid tumors. The combination of CAR-T cell technology with epigenetic modulation may thus represent the next generation of immuno-therapeutics.

## Figures and Tables

**Figure 1 cancers-15-01935-f001:**
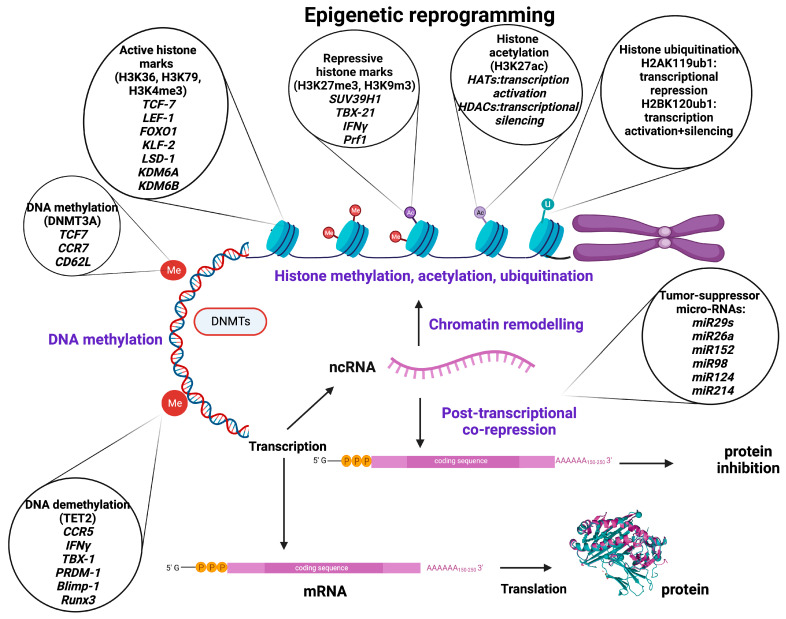
Basic epigenetic mechanisms, major epigenetic regulators, and target molecules involved in the regulation of genes related to T-cell stemness, memory, exhaustion, and trafficking as well as tumor immunity, development, and growth. *Created with BioRender.com.*

**Figure 2 cancers-15-01935-f002:**
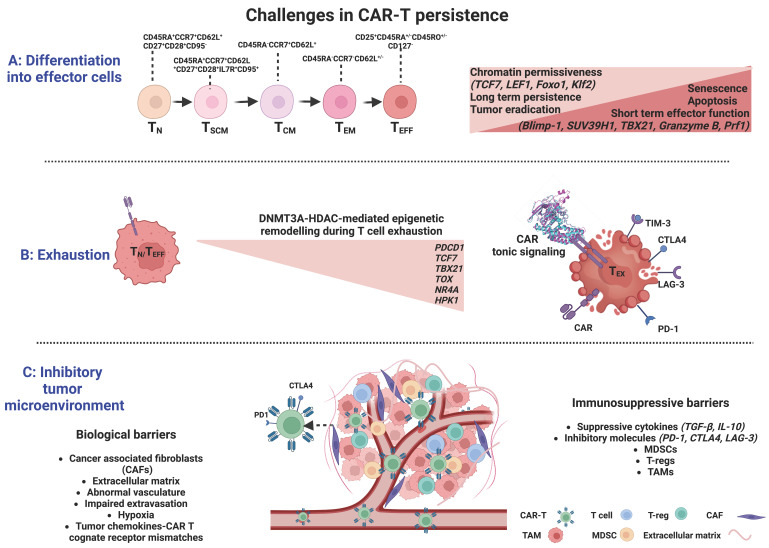
Significant barriers limiting CAR-T cell efficacy and their complex interplay with epigenetic regulation. (**A**) Differentiation into effector cells is associated with acquisition of repressive methylation marks at naïve cell-associated genes and demethylation marks at defined effector molecules. (**B**). The effector-to-exhaustion transition of T cells is DNMT3A- and HDAC-mediated and characterized by the sustained expression of several inhibitory receptors (*PD-1*, *CTLA-4*, *LAG3*, and *TIM3).* Exhaustion may be induced by antigen-independent receptor signaling and dimerization, resulting in aggregation of CARs and auto-activation (*tonic signaling*). (**C**). The many biological and immuno-suppressive barriers within the tumor constitute a complex and hostile tumor microenvironment, allowing uncontrolled proliferation and immune escape of cancer cells, preventing CAR-T cell infiltration, and driving CAR-T cell dysfunction to ultimately inhibit the efficacy of T-cell immunotherapies (*CAF: cancer-associated fibroblast*; *TAM: tumor-associated macrophage*; *MDSCs: myeloid-derived suppressor cells*; *Tregs: regulatory T cells) Created with BioRender.com.*

**Table 1 cancers-15-01935-t001:** Major epigenetic regulators associated with T-cell differentiation, exhaustion, and infiltration.

Epigenetic Reprogramming Strategies to Overcome CAR-Ts Roadblocks	Targets	Modification	Function on T Cells	Reference
Promoting stemness	CCR7	Demethylation	Promoting the dedifferentiation of effector into memory cells	[65]
	TCF7			
	TET2	DNA Methylation—KO	Promoting memory T-cell differentiation	[66]
	Il2ra	Histone H3-acetylation and reduced histone H3K9-trimethylation	Improving memory cell formation and anti-tumor activity	[69]
		miR150	Reduces effector function and proliferation	[83,84]
	SUV39H1	H3K9-trimethylation	Increasing long-term memory reprogramming capacity	[71,85]
	PDCD1	DNA methylation	Inhibits naïve to effector CD8 T-cell differentiation	[86]
	TCF7	DNA methylation	Maintaining of naïve and memory T-cell states	[76]
	LEF1			
	PRDM1	DNA methylation—KO	Avoiding the maintenance of T_EFF_ cell states	[77]
	TBX21			
	c-Myb	Non-coding RNA-mediated mechanisms by miR-150 (absence of miR-150)	Enhancing CD8^+^ T-cell memory differentiation	[81]
	PTEN	miR-214	Enhances proliferation	[87]
	Glut-1	miR-143	Promotes memory development	[88]
	PRDM1	miR-23a	Reduces T-cell differentiation and effector function	[89]
	NF-κB	miR-146a	Regulates and reduces effector function	[90]
	ARRB2	miR-150	Reduces effector function and proliferation	[83,84]
	Bcl-2	miR-15/16	Inhibits memory T-cell formation and differentiation	[91]
	Pim-1			
	Il7r			
	CD28			
	EOMES	miR139–342	Reduce effector function and differentiation	[83]
	Perforin			
	PDL-1	miR-873	Attenuates stemness and resistance of tumor cells	[92]
Promoting stemness	CREB-1	miR-17	Restrains i-Treg differentiation	[93]
	Runx3	Histone deacetylation	Inhibits differentiation into cytotoxic effector cells	[94]
	PRDM1			
Overcoming exhaustion	PDCD1	shRNA-mediated knockdown	Enhancing the anti-tumor efficacy of CLL-1-, mesothelin-, EGFR-, CD19-, and GPC3- CAR-Ts	[95,96,97,98]
		DNA methylation	Reverses exhaustion	[86]
	HPK1	Genetic depletion or pharmacological inhibition	Improving the exhaustion of CAR-Ts and the immune responses	[99]
	TOX-bound HBO1 complex	Histone H3 and H4 deacetylation	Reversing exhaustion	[100]
	PD-1	shRNA-mediated knockdown	Enhance the secretion of IFN-γ and the resistance to apoptosis	[101]
	TIM-3			
	LAG-3			
	CTLA-4	miR-28	Reduces exhausted T cells and regulates the cytokine secretion in the tumor microenvironment	[102]
	PD-1			
Promoting infiltration	ΕΖH-2	H3K27 trimethylation and DNA methylation	Preventing CXCR3 + Th1 cell infiltration in the TME	[103]
	CCR2	Let-7 miRNA	Impairs trafficking	[104]
	CCR5			
	SHIP-1	miR-155	Enhances trafficking and function of CD8^+^T cells	[105,106,107]
	SOCS-1			
	PHLPP2	miR-19–92	Enhances IFN-γ release and reduces inhibition of proliferation	[108]
	PTEN			
	DUSP5	miR-181a	Augments the sensitivity to peptide antigens and induces tolerance	[109]
	DUSP6			
	PTPN11			
	PTPN22			
	TNFα	miR-181a/b	Reduces cytotoxicity	[110]
	IDO1	miR-448	Enhance proliferation and antitumor function of CD8^+^ T cells	[111]
		miR-153		[112]
	VEGF-A	miR-126	Reduces cell proliferation, inhibits angiogenesis	[113,114,115]
	COX-2	miR-137	Contributes to the upregulation of retinoblastoma cell proliferation and invasion	[116]
	VEGF-A	miR-503-5p	Inhibits angiogenesis	[117]
	CXCL-1	miR-141	Inhibits Tregs recruitment	[118]
	Galectin-9	miR-22	Suppresses cell growth, invasion, and metastasis	[119]
Promoting infiltration	CD73	miR-30a-5p	Inhibits cell proliferation, cell migration and invasion	[120]
	CD73	miR-422a	Inhibits adenosine production	[121]
	COX-2	miR-708	Decreases proliferation, survival, and migration of lung cancer cells	[122]
	mPGES-1			

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
