# Peer review of "Empowering the Potential of CAR-T Cell Immunotherapies by Epigenetic Reprogramming"

_cancers, 2023, doi:10.3390/cancers15071935_

Round 1
Reviewer 1 Report
In this review the authors highlight the epigenetic basis of reduced persistence and efficacy of CAR T cell therapy in solid TME and further discuss the evidence and rationale to correct or modify these epigenetic changes in CAR T cells to augment their function by modulating their trafficking, differentiation, exhaustion, persistence, and effector function.
Overall this is a really good review on clinically important topic, and the author did a good job to provide evidences that support the critical role that epigenetic modifiers play in shaping the overall tumor-infiltrated T cells in general and CAR T cells in particular.
Below are specific comments addressing them may help improve overall quality and readability of the review.
1. T cell metabolism underlie the phenotype, effector function, and memory differentiation of T cells under homeostasis and TME, and similar metabolic mechanisms might play a decisive role in CAR T cell function and persistence. It would be great if the authors include a short discussion on interplay between metabolic and epigenetic programming of CAR T cells in TME. in other words, is there any evidence exist that suggest the potential metabolic modulation upon epigenic reprogramming.
2. Most of the immunotherapy recipients including CAR T cells seems to be late mid-age and older people, which experience multiple age-related systemic changes that might negatively influence the persistence of CAR T cells, via senescence and inflammaging-related mechanisms. It would also be helpful if the authors discuss such issues and any potential help that epigenetic modulation might play to circumvent such issues.
3. I encourage authors to include new papers that cover epigenetic mechanisms of CAR T cell phenotype and function (Akbari B et al. Cancer Lett. 2022; Ito Y and Kagoya Y, Cancer Sci, 2022; Belk JA et al. Cancer Cell, 2022; Garcia-Prieto CA et al. J Natl Cancer Inst. 2022; Lei X et al. Clin Cancer Res. 2021).
4. A reference is needed for the sentence "By assessing the epigenetic profile of T cells.......CD4+ and CD8+ T cell fractions, form-533 ing an anti-tumor phenotype".
Author Response
- T cell metabolism underlie the phenotype, effector function, and memory differentiation of T cells under homeostasis and TME, and similar metabolic mechanisms might play a decisive role in CAR T cell function and persistence. It would be great if the authors include a short discussion on interplay between metabolic and epigenetic programming of CAR T cells in TME. in other words, is there any evidence exist that suggest the potential metabolic modulation upon epigenic reprogramming.
Response.
We appreciate the reviewer’s comment. In the revised manuscript, we have added the subsection “Metabolic modulation upon epigenetic reprogramming” within the section “Tumor microenvironment’’ which reads as:
‘’3.3.2. Cell metabolism and epigenetics
Cellular metabolism plays a central role both in cancer cell growth and proliferation as well as in T cell fitness and potency and anti-tumor responses are potentially regulated by metabolic reprogramming. An interplay of metabolic events between T cells and the TME is taking place in the context of cancer where metabolic and nutrient changes in TME reshape the metabolic status of T cells impacting their activation, differentiation and exhaustion while generate an immunosuppressive milieu. On the other hand, the cell metabolome is interrelated to epigenome and epigenetic modifications in cancer are strongly linked to cellular metabolism by controlling expression of enzymes involved in metabolic pathways while metabolism affects epigenetic regulation through biosynthesis of macromolecules and energy production[178-180]. These interactions are bidirectional and synergistic in cancer. For instance, histone lysine methyltransferase SETD2, involved in complex different histone modifications, links epigenetic reprogramming with “metabolic memory” in prostate cancer by contributing to EZH2 and AMPK signaling pathway[178]. The c-Myc oncogene-mediated inhibition of succinate dehydrogenase complex subunit A (SDHA) via acetylation and activation of deacetylase degradation pathways, leads to cellular succinate accumulation further triggering H3K4me3 activation, tumor-specific gene expression and, thus, tumor progression[179]. In addition, tumors with H3.3K27M mutation could promote glutamine and glucose metabolism, leading to an epigenetic status marked by H3K27me3 deficiency while the genetic (e.g., shRNAs) or pharmacological interruption of these metabolic/epigenetic pathways results in suppressed H3.3K27M cell growth in vitro and tumor progression in vivo[180,181].
Metabolic events involve a variety of metabolites, such as acetyl-CoA, nicotinamide adenine dinucleotide (NAD+), S-adenosyl methionine (SAM), ATP, Flavin adenine dinucleotide (FAD), succinate and α-ketoglutarate (αKG). These molecules by dynamically regulating the metabolic status of DNA and histones play essential roles (as substrates or cofactors) in epigenome control during cancer development ([182] and references within).
Apart from the effect of metabolome/epigenome on the tumor growth, their interplay has also important consequences for T cell differentiation, function and fate[183,184]. Noumerous, external (ie oncometabolites, cytokines) or internal metabolic factors (ie acetyl-CoA, SAM) can influence the epigenetic landscape of T cells whereas epigenetic reprogramming directly influences T cell metabolism by affecting regulatory enzymes involved in glycolysis, OxPhos or mitochondrial biogenesis ([185]and references within). The broad use of multi-omics shed light on the interrelationship between epigenetics and metabolism and cultivated the idea of manipulating T cell metabolism via epigenetic modifications towards improving cancer immunotherapy. The enhanced capacity for glycolysis of terminally differentiated CD8+CD28− memory T cells, an accumulating population during human aging, has been linked to downregulation of Sirtuin1` (SIRT1)/FoxO1 axis; empowering the SIRT1–FoxO1 axis has been proposed as a targeted intervention for reprogramming terminally differentiated memory T cells and delaying the immune aging process[186]. Pharmacologic inhibition of Sirtuin-2 (Sirt2), a NAD+-dependent deacetylase inhibiting T cell metabolism and impairing T cell effector functions, endowed human tumor-infiltrating lymphocytes (TILs) with superior metabolic fitness and effector functions. [187] Protein arginine methyltransferase 5 (PRMT5) has a great impact in T cell differentiation and function[188] through the induction of cholesterol biosynthesis while specific Prmt5 deletion in CD4+ T cells suppressed Th17 cell differentiation, and protected mice developing experimental autoimmune encephalomyelitis [189]. The reported metabolic exhaustion of TILs has been attributed to direct competition between TILs and cancer cells for metabolic resources and subsequent environmental stress-induced epigenome remodeling within TILs, resulting from the loss of histone methyltransferase EZH2. In contrast, reprogrammed T cells expressing a gain-of-function EZH2 mutant displayed an enhanced ability to inhibit tumor growth in vitro and in vivo, thus suggesting that manipulation of T-cell EZH2 in cellular therapies may provide cellular products able to withstand solid tumor metabolic–deficient environments[190,191]. The metabolic reprogramming of effector CD8+ T cells through protein kinases MEK1/2 inhibition enhanced mitochondrial biogenesis and fatty acid oxidation and induced TSCM with increased self-renewability, multipotency and proliferative capacity, less exhaustion and strong antitumor effects in vivo. [192,193].
MiRNAs also affect T cell metabolism. Forced expression of miR-155 in tumor antigen-specific T cells improved tumor control and the miR-155-transduced T cells exhibited increased proliferation and effector functions associated with higher glycolytic activity independent of exogenous glucose, thus implicating that miR-155 may optimize the antitumor activity of adoptively transferred TILs by rendering them more resistant to the glucose-deprived environment of solid tumors.[113,199-201]. miR-143 overexpression enhanced the specific killing of HER2-CAR T cells against esophageal cancer by promoting memory T cell differentiation and metabolism reprogramming (T cell glucose uptake- and glycolysis inhibition) through Glucose transporter 1 (Glut-1). miR-143 expression and the regulation on T cell differentiation are suppressed by IDO and its metabolite, kynurenine, in the tumor microenvironment, thus IDO inhibition in the TME might increase the expression of miR-143 and enhance the antitumor effects of T cells by promoting Tm cell differentiation [95]. Decreased miR-34a expression in hypoxic tumor environments has been correlated with increased lactate concentration and increased LDHA in lactate-abundant tumors, resulting in impaired T-cell immune surveillance, pointing out T cell LDHA and miR34a as potential therapeutic targets for improved adoptive immunotherapy.[196].’’
Lines 652-869
- Most of the immunotherapy recipients including CAR T cells seems to be late mid-age and older people, which experience multiple age-related systemic changes that might negatively influence the persistence of CAR T cells, via senescence and inflammaging-related mechanisms. It would also be helpful if the authors discuss such issues and any potential help that epigenetic modulation might play to circumvent such issues.
Response.
We greatly appreciate the reviewer’s comment. We have now commented on senescence and inflammaging-related mechanisms in conclusion
‘’Recently, the role of T cells as accelerators of inflammaging, a state of chronic, low level inflammation in the elderly characterized by an increase in the levels of pro-inflammatory molecules in blood and tissues and loss of protective immunity as a result of genetic, environmental or stochastic factors has been recognized[214,215]. Changes introduced by such factors and possibly contributing to inflammaging, are molecularly recorded by epigenetic modifications, being heritable to subsequent cell generations and leading to accumulation of cell- and immune senescence [216]. Given the high numbers of late mid-age and older patients who are candidates for CAR T cell therapy but experience multiple age-related conditions, a number of epigenetic modulation strategies and senolytic CAR T-cells (targeting senescence-specific antigens) have recently been developed to optimize CAR-T cell fitness for older patients and broaden CAR T cell therapy to aging-related diseases[as reviewed in 223].
Nevertheless, the field is new, the epigenetic regulation pleiotropic and consequently, the intended CAR T cell epigenetic remodeling highly complex. Undoubtdedly, epigenome editing integrated in the CAR T manufacturing process will add to the production complexity and thereby, the already high cost. In addition, epigenome editing technologies, although not relying on the alteration of the underlying DNA and thus considered presumably safer, should be highly faithful and extensively tested for as currently underappreciated on- and off- target consequences, given that the epigenetic regulators can affect multiple pathways within the cells and some of the candidate target genes (DNMT3A, PRDM1) are associated with tumorigenesis. ‘’
Lines 1133-1165
- I encourage authors to include new papers that cover epigenetic mechanisms of CAR T cell phenotype and function (Akbari B et al. Cancer Lett. 2022; Ito Y and Kagoya Y, Cancer Sci, 2022; Belk JA et al. Cancer Cell, 2022; Garcia-Prieto CA et al. J Natl Cancer Inst. 2022; Lei X et al. Clin Cancer Res. 2021).
Response.
We thank the reviewer for this comment and appreciate the suggestion of including these papers (except Lei X et al)
Ref 20, 77, 151, 79 respectively
- A reference is needed for the sentence "By assessing the epigenetic profile of T cells.......CD4+ and CD8+ T cell fractions, form-533 ing an anti-tumor phenotype".
Response.
We thank the reviewer for this point and added the appropriate reference in the revised manuscript
Ref 169

Reviewer 2 Report
The topic of the review fulfills the aims of this special issue. Epigenetic reprogramming indeed represents the present and the future in the CAR T cell therapy field. The authors touched on all the main topics and exhaustively updated the state of the art on CAR T limitations and epigenetic reprogramming of CAR T cells.
The sections are well organized and well written and this has allowed an easily read of the review.
Figures and tables are clear, maybe I would increase the font size in the figures.
Minor comments:
- typo error in lines 392 (decision) and 434 (clinical)
Author Response
Response.
We appreciate the reviewer’s comments. In the revised manuscript, we increased the font size in the figures and corrected the typo errors, as suggested.
